# Design of a SIMO Deep Learning-Based Chaos Shift Keying (DLCSK) Communication System

**DOI:** 10.3390/s22010333

**Published:** 2022-01-03

**Authors:** Majid Mobini, Georges Kaddoum, Marijan Herceg

**Affiliations:** 1Department of Electrical and Computer Engineering, Babol Noshirvani University of Technology, Babol 47148-71167, Iran; 2Département de Génie Électrique, University of Québec, École de Technologie Supérieure, Montréal, QC H3C1K3, Canada; georges.kaddoum@etsmtl.ca; 3Faculty of Electrical Engineering, Computer Science and Information Technology Osijek, Josip Juraj Strossmayer University of Osijek, 31000 Osijek, Croatia; marijan.herceg@ferit.hr

**Keywords:** chaos shift keying, deep learning, LSTM, multi-antenna

## Abstract

This paper brings forward a Deep Learning (DL)-based Chaos Shift Keying (DLCSK) demodulation scheme to promote the capabilities of existing chaos-based wireless communication systems. In coherent Chaos Shift Keying (CSK) schemes, we need synchronization of chaotic sequences, which is still practically impossible in a disturbing environment. Moreover, the conventional Differential Chaos Shift Keying (DCSK) scheme has a drawback, that for each bit, half of the bit duration is spent sending non-information bearing reference samples. To deal with this drawback, a Long Short-Term Memory (LSTM)-based receiver is trained offline, using chaotic maps through a finite number of channel realizations, and then used for classifying online modulated signals. We presented that the proposed receiver can learn different chaotic maps and estimate channels implicitly, and then retrieves the transmitted messages without any need for chaos synchronization or reference signal transmissions. Simulation results for both the AWGN and Rayleigh fading channels show a remarkable BER performance improvement compared to the conventional DCSK scheme. The proposed DLCSK system will provide opportunities for a new class of receivers by leveraging the advantages of DL, such as effective serial and parallel connectivity. A Single Input Multiple Output (SIMO) architecture of the DLCSK receiver with excellent reliability is introduced to show its capabilities. The SIMO DLCSK benefits from a DL-based channel estimation approach, which makes this architecture simpler and more efficient for applications where channel estimation is problematic, such as massive MIMO, mmWave, and cloud-based communication systems.

## 1. Introduction

Chaotic signals are wide-band noise-like signals with robust and reproducible statistical features [1,2]. Thus, digital modulation using chaotic signals presents an inherently simple solution for robust and secure communications over multi-path fading channels [3,4]. Chaos-based modulations are generally classified into two main classes [5]. The first is coherent detection schemes, such as the Chaos Shift Keying (CSK) scheme [6], in which data are transmitted in a combination of basis functions obtained from chaotic waveforms. Chaotic synchronization is commonly used in the recovery of basis functions on the receiver side [7,8,9]. Theoretically, CSK modulation can achieve the Bit Error Rate (BER) performance of Binary Phase Shift Keying (BPSK) [10]. In application, this performance is not achievable as some problems needs to be solved, namely the recovery of basis functions and the estimation [11,12,13,14,15,16]. Since chaotic synchronization is still practically impossible in a turbulent environment [17], coherent schemes can not work properly in utilization.

The Differential Chaos Shift Keying (DCSK) system [18] is a variant of CSK where the basis functions have a special arrangement and the information can be revealed from the correlation between the parts of the basis functions. In DCSK modulation, each bit duration is divided into two equal slots, where the first slot is allocated to a reference signal whereas the second slot is used to send information-bearing signal. At the receiver side, these two signals are correlated to detect the transmitted bit information. Non-coherent schemes, such as DCSK, do not require Channel State Information (CSI) or chaotic synchronization on the receiver side [19,20,21]. The low data rate and energy efficiency of this scheme are significant drawbacks, as half of the bit time interval is used for sending non-information-bearing samples [22,23,24,25]. Moreover, sending the same signal twice negatively affects security [26].

The problem of basis function recovery from the received signals, independent of modulation, has impeded the progress of coherent chaos-based communications. To tackle this problem, we propose a Deep Learning (DL)-based receiver that enables us to implicitly estimate wireless channels and retrieve the transmitted message. We showed that using a trainable receiver can mitigate the chaotic synchronization problem in the existing correlator receivers.

Accepting the definitions of coherent and non-coherent chaos-based communications systems introduced in [8], the symbol may be recovered by means of coherent detection, where all possible sample functions are known, or by non-coherent detection, where one or more characteristics of the sample functions are estimated. Although in the proposed receiver, the sample functions are available, but the available information is previously estimated from the transmitted signals. Regarding the above definitions, though the proposed receiver has similarities to the coherent and non-coherent receivers, it does not exactly fit into the conventional categories. Hence, we use the term Deep Learning-based Chaos Shift Keying (“DLCSK”) to introduce a new category of chaos-based communications systems. Note that, a trained DLCSK receiver can easily work with existing CSK transmitters.

The proposed scheme shows a BER gain compared to conventional DCSK. This advantage is obtained at the cost of adding an offline training step. In other words, we train the receiver by chaotic reference signals, instead of reference signal transmission in the test phase. When the training process is performed under different channel conditions, NN grasps different chaotic maps and implicitly estimates channel distribution. As a result, DLCSK consumes fewer resources (energy and time) to transmit one bit and shows a more robust behavior.

### 1.1. Background

DL is a powerful tool that can be implemented for solving problems that are complicated to describe through mathematical models [27]. There are two general methods for designing DL-based architectures, namely DL-based receiver design and joint DL-based transceiver design [28]. A DL-based joint transceiver design optimizes the system as an end-to-end auto-encoder [29]. In contrast, a DL-based receiver optimizes one or more blocks in the receiver [30]. As an example of the latter, in [31], the authors suggested a DL-based receiver to indirectly estimate wireless channels and retrieve the signals. They demonstrated that DL-based receivers have the ability to learn the features of wireless channels, including non-linear distortions. Practical wireless channels change over time with a large dynamic range. In this case, NNs can hardly perform well in the detection tasks under different channel coefficients [32]. To overcome this challenge, Deep Transfer Learning (DTL) adopts a NN to extract the time-varying features with a few online training data by transferring knowledge from a source domain to a target domain [33,34,35]. Motivated by this approach, in this paper, we try to propose a framework for detection of the chaotic-modulated signals.

In this study, our focus is on the design of a DL-based receiver that can be easily harmonized with existing CSK transmitters. Recently, DL have attracted a lot of attention because of its effectiveness in data-driven analysis of chaotic dynamics [36]. Some studies have used DL in chaos-based communications [37,38,39,40], however, their methods need further explorations. In [39], the authors proposed an algorithm to demodulate the reference chaotic signals iteratively. In contrast to traditional demodulation methods, utilizing time or frequency resources to improve reliability, the iterative receiver addresses the feature extraction capability of neural networks (NNs). In [40], an intelligent OFDM-DCSK demodulator is proposed using an LSTM-aided NN that withdraw the correlations between chaotic modulated OFDM-DCSK signals in order to retrieve the transmitted information. However, the LSTM-aided receiver transmits reference signals that do not carry useful information. A similar problem can be observed in the other proposed DL-based modulations (e.g., in [38,39]).

In this paper, we present an innovative Long Short-Term Memory (LSTM)-based receiver that does not require any reference signal transmissions or chaotic synchronization circuits for data detection. The proposed DL-based Chaos Shift Keying (DLCSK) system benefits from other advantages of NNs, such as generalization and fast training. The multi-antenna design of the DLCSK receiver that, uses an LSTM for each antenna is also presented. Many articles have also combined Multi-Antenna technology with Chaos-based communications [41,42,43,44,45,46,47,48,49,50,51,52,53]. For instance, a MIMO STBC-DCSK system does not require any complicated channel estimation, carrier synchronization, or rake reception [54]. However, these conventional techniques transmit the reference signals over the channel, which increases the redundancy and complexity of the system. Previous research in the context of neural network ensembles shows that combining network architectures is frequently more accurate than using single networks [55,56,57]. The proposed Single Input Multi Output (SIMO) DLCSK focuses on a fusing method to achieve a diversity gain, where the outputs of all LSTMs are combined using a majority voting strategy. This design maintains the advantages of traditional Multi-Antenna and Chaos-based systems, i.e., it does not require rake receivers, CSI, and chaotic synchronization. DLCSK is an appealing candidate for secure data transmission in cloud-based systems [58,59,60], vehicular communications [61,62,63], and massive MIMO systems [64,65,66,67] due to the aforementioned characteristics.

### 1.2. Contributions

The objective of this work is to develop a DL-based receiver that benefits from the inherent security of chaotic signals and the merits of NNs. This design will robust the capabilities of the existing chaos-based schemes, such as reliability and energy efficiency. Innovative aspects of this paper are briefly defined below:We train a LSTM-based classifier that enables online classification of the received chaotic signals. Implementing this method can mitigate the chaotic synchronization problem in the existing CSK receivers;The DLCSK modulation/demodulation scheme does not need any reference transmission for basis functions recovery, unlike reference-based non-coherent modulations. According to the above advantages, the BER performance of a single antenna DLCSK is close to the performance of the antipodal CSK under a Rayleigh fading channel that shows an outstanding BER performance among all chaotic modulations [10].We have proposed a Multi-Antenna architecture of the DLCSK system. Multiple LSTMs are used at the receiver end to obtain a diversity gain. We numerically simulate the SIMO DLCSK structure and state the advantages of fusing the hard outputs of the LSTMs to come to a decision.

The rest of this paper is organized in following sections:□In Section 2, a statistical study on the existing CSK systems and correlation receivers is presented;□In Section 3, the structure of the proposed SIMO DLCSK system and the basics of the LSTM-based classifier are described;□In Section 4, simulation results and discussions are explained;□In Section 5, the conclusions are explained.

## 2. Traditional Correlation Receivers

This section presents a statistical study on a typical CSK system equipped with the correlation receivers for a deeper identification of its unsolved problems. In contrast with regular communications systems in which the basis functions are periodic and orthogonal (e.g., sinusoidal functions), in a chaotic communications system, the basis functions are not orthogonal necessarily and differ from symbol to another. Consider now a coherent CSK system using two basis functions [10]. The elements of the signal set are given by
(1)$$s_{b}=s_{b1}\overset{˘}{x}+s_{b2}\tilde{x},$$
where *b* is the index of the current transmission bit, the weights $s_{b1}$ and $s_{b2}$ are the elements of the signal vector, and the basis functions $\overset{˘}{x}$ and $\tilde{x}$ are discrete-time chaotic signals with β samples, i.e., $\overset{˘}{x}$$=\{{\overset{˘}{x}}_{1},\cdots ,{\overset{˘}{x}}_{\beta }\}$ and $\tilde{x}$$=\{{\tilde{x}}_{1},\cdots ,{\tilde{x}}_{\beta }\}$. A binary bit data symbol is spread by a chaotic signal with the bit duration $T_{b}$. Thus, we have $T_{b}=\beta T_{c}$, where $T_{c}$ is the time between each chaotic sample (chip). The transmitted sample functions are $s_{1}=\sqrt{E_{b}}\;\overset{˘}{x}$ and $s_{2}=\sqrt{E_{b}}\;\tilde{x}$, displaying symbols “0” and “1”, respectively. The corresponding signal vectors are ($s_{11}$$s_{12}$)=($\sqrt{E_{b}}\;0$) and ($s_{21}$$s_{22}$)=(0 $\sqrt{E_{b}}$), where $E_{b}$ stands for the average energy per bit. With two basis functions, the receiver should be configured with at least two correlators. The message is detected by correlating the received signal with two reference signals ${\overset{˘}{x}}^{\prime }$ and
${\tilde{x}}^{\prime }$, and forming the corresponding observation signals (so-called decision variables) $D_{b1}$ and $D_{b2}$. If $D_{b1}>D_{b2}$, then the decision is “1”, and if $D_{b1}<D_{b2}$, then the decision is “0”. Consider now the output of a correlator under a noisy channel. In a coherent CSK system, the reference signals ${\overset{˘}{x}}^{\prime }$ and ${\tilde{x}}^{\prime }$ are derived from the noisy received signal ($s_{b}+n$). The decision variable $D_{b1}$, at the output of the correlator, can be written as
(2)$$\begin{array}{cc} D_{b1} & =\int_{T_{s}}^{T}\left[s_{b}+n\right]{\overset{˘}{x}}^{\prime }\;dt\\ \; & =\int_{T_{s}}^{T}\left[s_{b1}\overset{˘}{x}+s_{b2}\tilde{x}+n\right]{\overset{˘}{x}}^{\prime }\;dt\\ \; & =s_{b1}\int_{T_{s}}^{T}\overset{˘}{x}{\overset{˘}{x}}^{\prime }\;dt+s_{b2}\int_{T_{s}}^{T}\tilde{x}{\overset{˘}{x}}^{\prime }\;dt\\ \; & +\int_{T_{s}}^{T}n{\overset{˘}{x}}^{\prime }\;dt, \end{array}$$
where $T_{s}$ is the synchronization transient time for each bit duration. Note that $D_{b1}$ is a random variable, whose mean value depends on the $E_{b}$ and the quality of the regenerated reference signal ${\overset{˘}{x}}^{\prime }$. If a perfect synchronization is maintained throughout the transmission, we have $T_{s}=0$, ${\overset{˘}{x}}^{\prime }=\overset{˘}{x}$, and ${\tilde{x}}^{\prime }=\tilde{x}$. In this case
(3)$$\begin{array}{cc} D_{b1} & =s_{b1}\int_{0}^{T_{b}}\overset{˘}{x}\overset{˘}{x}\;dt+s_{b2}\int_{0}^{T_{b}}\tilde{x}\overset{˘}{x}\;dt\\ \; & +\int_{0}^{T_{b}}n\overset{˘}{x}\;dt. \end{array}$$

The chaotic basis function $\overset{˘}{x}$ is different in every transmission, and the variance of $\int_{0}^{T_{b}}{\overset{˘}{x}}^{2}\;dt$ causes detection error. This variance can be reduced by increasing the bit duration $T_{b}$. Alternatively, it can be zeroed by modifying the generated basis functions, such that the transmitted energy $E_{b}$ is kept constant. A constant $E_{b}$ is achievable by normalizing the basis functions before each symbol transmission, such that $\int_{0}^{T_{b}}{\overset{˘}{x}}^{2}\;dt=1$.

The second term on the right-hand side of Equation (Equation 3) results in the cross-correlation estimation problem. It can also be reduced by selecting long chaotic signals or using Walsh functions [9]. By choosing orthonormal basis functions, the estimated symbol $s_{b1}$ can be written as
(4)$$\begin{array}{cc} D_{b1} & =s_{b1}+\int_{0}^{T}n\overset{˘}{x}\;dt. \end{array}$$

In contrast with the traditional demodulation methods that often use synchronization (coherent schemes) or delayed reference transmission (non-coherent schemes) for reference recovery, the proposed DLCK utilizes DL for training and recovery of the reference signals ${\overset{˘}{x}}^{\prime }$ and ${\tilde{x}}^{\prime }$. By using DL, it not only mitigates the need for chaotic synchronization during transmissions, but also can reduce the effect of the last term in Equation (Equation 4) using an indirect channel estimation method. It should be noted that our main focus is not on the auto and cross-correlation problems, or selecting proper chaotic maps. Therefore, we have chosen the Chebyshev map and logistic map to make chaotic sequences simpler in terms of generating.

## 3. SIMO DLCSK System Model

For the general architecture of the SIMO DLCSK system, refer to Figure 1. The proposed design involves two phases. In the first phase, the chaotic sequences are transmitted through an offline training phase under different channel conditions, and then the deployment/test phase initializes the modulated data transmission.

### 3.1. Chaotic Signals

We consider discrete-time chaotic signals generated by the Chebyshev map and the Logistic map for “0” and “1”, respectively. The chaotic Chebyshev map generated by the second order Chebyshev polynomial function is used, which can be written as ${\overset{˘}{x}}_{t+1}=1-{\overset{˘}{x}}_{t}^{2}$ [68]. Another chaotic map is the Logistic map, which is generated using a recursive function, ${\tilde{x}}_{t+1}=\rho {\tilde{x}}_{t}\left(1-{\tilde{x}}_{t}\right)$ [69]. The parameter ρ is called the bifurcation parameter, and the values of interest for ρ are in the interval [1,4]. For 3.57≤ρ≤4, the generated sequence is non-periodic and non-converging [70].

In this study, we evaluate the effect of the changes in ρ on the classifier’s performance. The parameter changes can be intentional (to achieve higher accuracy), or unintentional (due to environmental conditions) [71]. When ρ<3.57, the generated signals by Logistic map show a periodic behavior and a classifier can easily separate it from a chaotic signal, (e.g., a signal generated by the Chebyshev map). There is a trade-off between the classification accuracy and the security. Since a periodic waveform can bring negative effect on the security, we select ρ>3.6 for a case that definitely guarantees the chaotic dynamics. The chaotic maps have zero mean and unit variance, i.e., $E\left({\overset{˘}{x}}_{t}\right)=E\left({\tilde{x}}_{t}\right)=0$ and $E\left({\overset{˘}{x}}_{t}^{2}\right)=E\left({\tilde{x}}_{t}^{2}\right)=1$, where E(·) denotes the probabilistic expectation operator.

### 3.2. Transmitter

The transmitter structure is similar to conventional coherent CSK transmitters. Thus, the proposed DLCSK demodulator can easily work with existing CSK transmitters. In the training phase, the transmitter generates two sets of the chaotic signals, ${\left\{{{\overset{˘}{x}}^{\prime }}^{\left(n\right)}\right\}}_{n=1}^{N}$ and ${\left\{{{\tilde{x}}^{\prime }}^{\left(n\right)}\right\}}_{n=1}^{N}$ using the Chebyshev and the Logistic maps, respectively. Each one of these sets includes the number of *N* chaotic signals with length β samples, i.e, ${\overset{˘}{x}{}^{\prime }}^{\left(n\right)}=\left\{\overset{˘}{x}{{}^{\prime }}_{1}^{\left(n\right)},\cdots ,\overset{˘}{x}{{}^{\prime }}_{\beta }^{\left(n\right)}\right\}$ and ${\tilde{x}{}^{\prime }}^{\left(n\right)}=\left\{\tilde{x}{{}^{\prime }}_{1}^{\left(n\right)},\cdots ,\tilde{x}{{}^{\prime }}_{\beta }^{\left(n\right)}\right\}$.

In other words, we use two sets of the chaotic signals, each with length S=Nβ samples, where *N* and β are the number and the length of the chaotic signals, respectively. All of the chaotic signals ${{\overset{˘}{x}}^{\prime }}^{\left(n\right)}$ and ${{\tilde{x}}^{\prime }}^{\left(n\right)}$ with known binary labels $j^{\left(n\right)}\in \left\{0,1\right\}$ are transmitted repeatedly to train the receivers. All the *M* antennas obtain the corresponding altered signals, i.e., ${\overset{˘}{r}}_{m}^{\left(n\right)}=\left\{{\overset{˘}{r}}_{m,1}^{\left(n\right)},\cdots ,{\overset{˘}{r}}_{m,\beta }^{\left(n\right)}\right\}$ and ${\tilde{r}}_{m}^{\left(n\right)}=\left\{{\tilde{r}}_{m,1}^{\left(n\right)},\cdots ,{\tilde{r}}_{m,\beta }^{\left(n\right)}\right\}$, for input to the NN. After the number of 2N transmissions, we have a set of received vectors ${\left\{r_{m}^{\left(n\right)}\right\}}_{n=1}^{2N}=\{{\overset{˘}{r}}_{m}^{\left(n\right)}$, ${\tilde{r}}_{m}^{\left(n\right)}{\}}_{n=1}^{N}$ with known class labels ${\left\{j^{\left(n\right)}\right\}}_{n=1}^{2N}={\left\{0^{\left(n\right)},1^{\left(n\right)}\right\}}_{n=1}^{N}$ to train the NN through the supervised learning framework. Therefore, training set of the mth DLCSK receiver can be expressed as
(5)$$D_{m}={\left\{r_{m}^{\left(n\right)},j^{\left(n\right)}\right\}}_{n=1}^{2N},$$
where 2N is the number of chaotic signals in the training set. Assuming a Rayleigh fading channel, the $r_{m,t}$ (tth sample of the vector $r_{m}$) can be modeled as a complex-valued random variable, i.e., $r_{m,t}=ℜ\left(r_{m,t}\right)+ℑ\left(r_{m,t}\right)$, where the operators ℜ(.) and ℑ(.) represent the real and imaginary parts of a complex number, respectively. Therefore, the vector $r_{m}$ can be separated into two vectors, i.e., $r_{m}$ = $\left[ℜ\left(r_{m}\right),ℑ\left(r_{m}\right)\right]$, before being fed into the classifier, and the training set can be rewritten as
(6)$$D_{m}={\left\{{\left[ℜ\left(r_{m}\right),ℑ\left(r_{m}\right)\right]}^{\left(n\right)},j^{\left(n\right)}\right\}}_{n=1}^{2N}.$$

Once the DL-based receiver is trained, it can be used for online demodulation and data detection. In the test/deployment phase, the DLCSK modulator maps a transmission bit $b^{\left(z\right)}\in \left\{0,1\right\}$, (1≤z≤Z), to a chaos waveform, where $b^{\left(z\right)}$ denotes $z^{th}$ transmission bit and *Z* is the number of data bits. Thus, in the test phase, the transmitter generates two sets of the chaotic signals, i.e., ${\left\{{\overset{˘}{x}}^{\left(z\right)}\right\}}_{z=1}^{Z}$ and ${\left\{{\tilde{x}}^{\left(z\right)}\right\}}_{z=1}^{Z}$ using the Chebyshev and the Logistic maps, respectively. Each of these sets includes the number of *Z* chaotic signals with length β samples, i.e., ${\overset{˘}{x}}^{\left(z\right)}=\left\{{\overset{˘}{x}}_{1}^{\left(z\right)},\cdots ,{\overset{˘}{x}}_{\beta }^{\left(z\right)}\right\}$ and ${\tilde{x}}^{\left(z\right)}=\left\{{\tilde{x}}_{1}^{\left(z\right)},\cdots ,{\tilde{x}}_{\beta }^{\left(z\right)}\right\}$. According to the Equation (1), and depending on the current bit $b^{\left(z\right)}$, the signal $s_{b}^{\left(z\right)}$ is transmitted, i.e.,
(7)$$\begin{array}{c} s_{b}^{\left(z\right)}=\left\{\begin{array}{cc} \sqrt{E_{b}}\;{\overset{˘}{x}}^{\left(z\right)} & \;\mathrm{if}\;b^{\left(z\right)}=0\\ \sqrt{E_{b}}\;{\tilde{x}}^{\left(z\right)} & \;\mathrm{if}\;b^{\left(z\right)}=1. \end{array}\right. \end{array}$$

Since in every practical communication system, the chaos generator circuits may operate under different environmental conditions, it is essential to consider a parameter mismatch between the training and testing phases. To evaluate the system’s generalization capability and robustness, we use different parameter settings to achieve ${\left\{{{\overset{˘}{x}}^{\prime }}^{\left(n\right)}\right\}}_{n=1}^{N}\neq {\left\{{\overset{˘}{x}}^{\left(z\right)}\right\}}_{z=1}^{Z}$ and ${\left\{{{\tilde{x}}^{\prime }}^{\left(n\right)}\right\}}_{n=1}^{N}\neq {\left\{{\tilde{x}}^{\left(z\right)}\right\}}_{z=1}^{Z}$. Hence, to generate Logistic maps, different bifurcation parameters are chosen as $\rho_{train}$ and $\rho_{test}$. In addition, to generate Chebyshev maps, different initial states are chosen for the training and the deployment phases, i.e., ${\overset{˘}{x}}_{1}^{\prime }\neq {\overset{˘}{x}}_{1}$.

The transmission filter (or pulse shaping filter) is commonly used in communication systems. This filter can take different forms, such as the Gaussian filter or raised-cosine filter [72]. In this paper, we consider a rectangular pulse of unit amplitude on $\left[0,T_{c}\right]$, where $T_{c}$ represents the chip time. The noise power can be restricted by a receiving filter at the receiver side. Note that, these filters are not our main focus.

### 3.3. Channel Model and Estimation

In an AWGN channel, a one-dimensional noisy version of the transmitted signal $s_{b}$ can be observed at the receiver side. The mth receiver obtains the altered signals $r_{m}$ and makes a decision about the transmitted bit. In addition to the Gaussian noise effect, many other stochastic phenomena may occur in a practical communication channel. Conventional channel estimation approaches are sensitive in terms of the quality of the pilot signals. One way of augmenting DL models is through the use of learning channel variations [73]. If the LSTM-based classifier is trained using a dataset that contains signals transmitted under different channel conditions, the classifier will be resilient to channel changes, eliminating the need for instantaneous channel estimation [74,75]. In this paper, we assume that the channel changes from one signal transmission to another, and the transmitted signal acts as a pilot that carries channel information. Thus, the NN can simultaneously obtain different chaotic maps and estimate the statistical distribution of the fading.

Consider a multi-path fading channel, with *L* independent paths, where the channel coefficients follow a Rayleigh distribution. Therefore, the probability density function (pdf) of the channel coefficient α can be given as [76],
(8)$$q\left(\alpha |\delta \right)=\frac{\alpha }{\delta^{2}}e^{\frac{-\alpha^{2}}{2\delta^{2}}},$$
where δ>0 is the scale parameter of the distribution representing the root mean square value of the received voltage signal. Considering a multi-antenna system with the number of *M* antennas, if the tth transmitted sample is shown by $s_{b,t}$, the tth received sample at the mth antenna can be modeled as
(9)$$r_{m,t}=\sum_{l=1}^{L}\alpha {}_{m,l}s_{b,t-\tau_{m,l}}+n_{m,t},\;m=1,2,\cdots ,M,$$
where $\alpha_{m,l}$ represents the channel gain of lth path between the transmit antenna and mth receive antenna, and $\tau_{m,l}$ is the delay of the lth path. *L* is the number of paths, and $n_{m,t}$ is independent noise at each antenna, which is assumed to be additive white Gaussian noise with zero mean and variance $N_{0}/2$.

Since we assume that the channel changes from one signal transmission to another, the channel gain $\alpha_{m,l}^{\left(n\right)}$ and training SNR $\sigma_{tr}^{\left(n\right)}$ changes after the nth channel realization. In particular, $\sigma_{tr}^{\left(n\right)}$ is a Gaussian random variable, such that $\sigma_{tr}^{\left(n\right)}\in \left[\sigma_{tr,min},\sigma_{tr,max}\right]$, where $\sigma_{tr,min}$ and $\sigma_{tr,max}$ are optional SNR values. The goal of training process is to train a NN with a complex-valued vector $r_{m}$. The input complex values are split into real and imaginary parts, i.e., $r_{m}$= $\left[ℜ\left(r_{m}\right),ℑ\left(r_{m}\right)\right]$, before being fed into the classifier. Therefore, we have two feature vectors containing the channel coefficients and stochastic phases. Later, a Softmax layer estimates probability vectors $p_{n,j}$ from input distribution, for the nth observation, where *j* shows the possible classes (i.e., j∈{1,2}), and optimizes the cross-entropy cost function in Equation (Equation 13). In the test phase, we use this trained network to detect unknown inputs.

### 3.4. Receiver

We consider multiple antennas and LSTMs at the receiver end to establish a SIMO design and to obtain a diversity gain. This architecture relies on an ensemble method to fuse several classifiers with the goal of increasing the classification accuracy. In the following, we introduce the structure of a single LSTM-based classifier. The proposed classifier has five base layers: Sequence input layer, LSTM/BiLSTM, fully connected layer (size 2), Softmax, and Classification layer. The sequence input layer is only used to fetch sequential input values of length 2β. The adopted LSTM cell (unit) is shown in Figure 2.

The forget gate $f_{t}$ determines how much of the current cell state should be forgotten, and the output gate $o_{t}$ controls which part of the information should be sent to the output. $C_{t-1}$ and $C_{t}$, respectively, show the state value of the memory unit at the previous step and current step. Then, $h_{t}{}_{-}{}_{1}$ and $h_{t}$ indicate the output of the previous and the current states, respectively, whereas $r_{t}$ and μ represents the current input and sigmoid function, respectively. Equation (10) illustrates the LSTM cell calculation process [77]. The forget gate decides which information will be remembered or forgotten based on the last hidden layer output $h_{t}{}_{-}{}_{1}$ and the current input $r_{t}$. The memory cell value $C_{t}$ is determined by the current value $C_{t}$, its own state $C_{t}{}_{-}{}_{1}$, input, and forget gates. The operator (∗) represents element-wise matrix multiplication, while (·) denotes point multiplication. *w* represents the weight and *b* is the bias parameter.
(10a)$$\begin{array}{c} i_{t}=\mu \;\left(w_{i}\cdot \left[h_{t-1},r_{t}\right]+b_{i}\right), \end{array}$$
(10b)$$\begin{array}{c} f_{t}=\mu \;\left(w_{f}\cdot \left[h_{t-1},r_{t}\right]+b_{f}\right), \end{array}$$
(10c)$$\begin{array}{c} C_{t}=\tanh\;\left(w_{c}\cdot \left[h_{t-1},r_{t}\right]+b_{c}\right), \end{array}$$
(10d)$$\begin{array}{c} C_{t}=f_{t}*C_{t-1}+i_{t}*C_{t}, \end{array}$$
(10e)$$\begin{array}{c} o_{t}=\mu \;\left(w_{o}\cdot \left[h_{t-1},r_{t}\right]+b_{o}\right), \end{array}$$
(10f)$$\begin{array}{c} h_{t}=o_{t}*\tanh\left(C_{t}\right). \end{array}$$

In this work, motivated by the features of LSTM, a Bidirectional (BiLSTM) arrangement is implemented for classification tasks. The hidden state of BiLSTM at times *t* can be calculated by the weighted sum of the forward hidden state $\vec{h_{t}}$ and the backward hidden state $\overset{\leftarrow }{h_{t}}$ as follows:
(11a)$$\begin{array}{c} \vec{h_{t}}=\mathrm{LSTM}\left(r_{t},\vec{h_{t-1}}\right), \end{array}$$
(11b)$$\begin{array}{c} \overset{\leftarrow }{h_{t}}=\mathrm{LSTM}\left(r_{t},\overset{\leftarrow }{h_{t-1}}\right), \end{array}$$
(11c)$$\begin{array}{c} U_{t}=w_{t}\vec{h_{t}}+v_{t}\overset{\leftarrow }{h_{t}}+b_{t}, \end{array}$$
where $w_{t}$ and $v_{t}$ denote the weights corresponding to $\vec{h_{t}}$ and $\overset{\leftarrow }{h_{t}}$, respectively. The number of hidden units indicates the number of BiLSTM units that need to be placed in the hidden layer of the network.

The fully connected layer receives the output of the BiLSTM layer in order to increase the stability of the output by performing more non-linear operations. There are two fully connected layers for two output classes. The Softmax layer is an activation function that calculates a probability for each sequence and sends results to the next layer. This layer contains two nodes, which is the same as the number of output classes. The utilized Softmax function can be written as [78],
(12)$$\gamma {\left(U_{t}\right)}_{j}=\frac{e^{{\left(U_{t}\right)}_{j}}}{\sum_{i\in I}e^{{\left(U_{t}\right)}_{i}}},$$
where $\gamma {\left(U_{t}\right)}_{j}=p_{n,j}$ is the probability that vector $U_{t}$ is a member of jth class (j∈{0,1}), and I={0,1} is a set of all possible classes. In other words, $p_{n,0}$ and $p_{n,1}$, (1≤n≤N), represents the probability that the transmitted chaotic signal is “0” or “1”, respectively.

The goal of the training process is to minimize the categorical cross-entropy loss function,
(13)$$L\left(\theta \right)=-\frac{1}{a_{t}}\sum_{t=1}^{a_{t}}\sum_{j=0}^{1}p_{n,j}^{\prime }\;\log\left(p_{n,j}\right),$$
where $a_{t}$ is the mini-batch size, θ indicates set of network parameters corresponding to the different layers, $p_{n,j}$ is the Softmax’s layer output probability for output class *j* and observation *n*, and $p_{n,j}^{\prime }\in \left\{0,1\right\}$ represents the binary indicator if class label *j* is the correct classification for observation *n*. A popular algorithm to obtain θ is the Stochastic Gradient Descent (SGD) method [79], which starts with a random initial value $\theta =\theta_{0}$, and iteratively updates θ as
(14)$$\theta_{k+1}=\theta_{k}-\eta \nabla \tilde{L}\left(\theta_{k}\right),$$
where η>0 is the learning rate, and $\tilde{L}\left(\theta_{k}\right)$ is an approximation of the loss function which is computed for a random mini-batch of training examples of size $a_{t}$ at each iteration. Through offline training, a network with optimized weights and biases that can be used for online signal demodulation is formed. The proposed training algorithm is summarized in Algorithm 1.
**Algorithm 1:** Training of mth receiver.**> At the Transmitter side:**1:     Input parameters: Number of generated signals (2N), Spread factor (β).2:     Generate two sets of the chaotic signals, each with length *N*, using the Chebyshev and the Logistic maps, i.e., ${\left\{{{\overset{˘}{x}}^{\prime }}^{\left(n\right)}\right\}}_{n=1}^{N}$ and ${\left\{{{\tilde{x}}^{\prime }}^{\left(n\right)}\right\}}_{n=1}^{N}$.3:     Normalize chaotic signals.4:     Transmit all generated signals over the channel:             for n=1:2N                   Generate a random $\sigma_{tr}^{\left(n\right)}$$\in [\sigma_{tr,min},\sigma_{tr,max}]$;                   Generate a random $\alpha_{m,l}^{\left(n\right)}$;                   Transmit nth symbol;             Endfor.**> At the receiver side:**5:      The mth antenna receives altered signals{r${}_{m}^{\left(n\right)}{\}}_{n=1}^{2N}=\{$$\overset{˘}{r}$${}_{m}^{\left(n\right)}$, $\tilde{r}$${}_{m}^{\left(n\right)}{\}}_{n=1}^{N}$ with known labels ${\left\{j^{\left(n\right)}\right\}}_{n=1}^{2N}={\left\{0^{\left(n\right)},1^{\left(n\right)}\right\}}_{n=1}^{N}$.6:      Separate $r_{m}^{\left(n\right)}$ into real and imaginary parts to form the training set:$D_{m}={\left\{{\left[ℜ\left\{r_{m}\right\},ℑ\left\{r_{m}\right\}\right]}^{\left(n\right)},j^{\left(n\right)}\right\}}_{n=1}^{2N}$.7:      Train mth NN-based receiver including:              - Sequence input layer;              - LSTM/BiLSTM layer;              - Fully connected layer;              - Softmax layer;              - Classification layer;8:      End of Training.

### 3.5. Decision Combining Rule

Reliable communication over multi-path channels highly depends on the condition of the paths, and the probability of deep fade. Spatial diversity is used in conventional coherent communication systems for combating the destructive effects of small-scale fading, and thereby for improving reliability. However, these receivers are very sensitive to the accuracy of the channel estimation process. Since the proposed classifier is trained under different channel conditions, the receiver does not require complex channel estimation techniques or soft data combining methods, such as Equal Gain Combining (EGC) and Maximal Ratio Combining (MRC) [80], for data detection. We can combine hard outputs of several classifiers to achieve a diversity gain through a simple decision rule.

Based on the received vector $r_{t,m}$, each of the LSTMs can produce a local decision and report this decision to a Fusion Center (FC), which makes the final decision. There are several methods of fusing the decisions of the classifiers, such as majority voting and ensemble averaging [81]. In this paper, having the binary-valued decisions, the FC applies the majority voting fusion rule to generate the ultimate decision. The class with the highest overall output is selected as the ultimate decision. Mathematically, the decision class O(r) can be calculated as [82]
(15)$$O\left(r\right)=\arg\;\max_{j}\;\sum_{m=1}^{M}{Ø}_{j}{,}_{m}\left(C_{m}\left(r\right)=j\right),$$
where *M* is the number of classifiers, j∈{0,1} denotes the jth class, $C_{m}\left(r\right)$ represents the output of the mth classifier for the received vector r, and ${Ø}_{j}{,}_{m}$ is a binary characteristic function that can be defined as
(16)$$\begin{array}{c} {Ø}_{j}{,}_{m}=\left\{\begin{array}{cc} 1 & \;\mathrm{if}\;C_{m}\left(r\right)=j\\ 0 & \;\mathrm{if}\;C_{m}\left(r\right)\neq j. \end{array}\right. \end{array}$$

## 4. Simulation Results

In this section, we first provide a comparison between the BER performance of the Single-input Single-output (SISO) DLCSK and the conventional DCSK over AWGN and multi-path Rayleigh fading channels. Then, evaluation of various parameters such as bifurcation parameters and the number of antennas on the system performance takes place. In all simulations the classifiers are trained only at a limited SNR range, i.e., $\sigma_{tr}^{\left(n\right)}\in [\sigma_{tr,min}$, $\sigma_{tr,max}]\;$ dB, while tested over a wide range of $E_{b}/N_{0}$ values greater than 0dB.

In order to generate chaotic signals, we choose two discrete time recursive functions, i.e., Chebyshev and Logistic maps, which have been used extensively in practical communication systems [40,83]. The transmitter creates two sets of chaotic signals using these two maps during the training phase, according to the values in Table 1. For example, in case 1, each of these sets includes the number of N=2000 chaotic signals, each with length β=50 samples. In other words, we use two sets of the training chaotic signals, each with total length $S=50\times 2000=10^{5}$ samples. All of the chaotic signals have equivalent class labels as j∈{0,1}. The receiver obtains the corresponding altered signals and forms the final training set as input of the NN. Therefore, after the number of 2 × 2000 = r 4000 signal transmissions, we will obtain a training set, consisting of 4000 received signals along with their corresponding class labels j. In the test phase, a binary “0” will be sent, transmitting a Chebyshev map, and if “1” is to be sent, a Logistic map is transmitted.

In our simulations, to evaluate the system’s generalization capability and robustness against parameter mismatch, we use different parameter settings for the training and testing phases. Therefore, to generate Logistic maps, the bifurcation parameters are, respectively, chosen as $\rho_{train}=3.6$ and $\rho_{test}=3.3$ for the training and the deployment phases, with an initial state as ${\tilde{x}}_{1}=0.3$. To generate Chebyshev maps, initial states are chosen as ${\overset{˘}{x}}_{1}=0.3535$ and ${\overset{˘}{x}}_{1}=0.3$ for the training and the deployment phases, respectively.

Predefined functions of the MATLAB Neural Network Toolbox can be used to define an LSTM network and specify training options, including Learning Rate (η), and the number of hidden layers (*H*). In all experiments, the learning rate is set to η=0.01. The other DL parameters are selected based on the SIMO DLCSK parameters. Although we consider the AWGN and Rayleigh fading channels, SIMO DLCSK can also be applied to any other channel model as well. The considered network parameters are listed in Table 1.

### 4.1. Training Convergence

Figure 3, illustrates the training metrics of a single classifier under the AWGN channel and how the training is performed for a random training SNR value $\sigma_{tr}^{\left(n\right)}\in \left[19,23\right]$ dB. Each epoch is a full pass through the dataset, and each iteration is an update of the network parameters. Figure 3 also shows validation metrics, which are recorded each time the program validates the network. For example, the validation accuracy presents the classification accuracy on the validation set, which gives an idea of the generalization of the model. Final validation metrics are labeled “Final” in the plot.

The training accuracy represents the classification accuracy of each mini-batch during the training process. The smoothed training accuracy is less noisy than the training accuracy and makes it easier to observe trends. The training loss shows the value of the loss function, i.e., the categorical cross-entropy, on each mini-batch. The results show that the loss function converges rapidly within about 100 iterations.

When the DLCSK system is trained at relatively lower SNRs (For instance, when $\sigma_{tr}^{\left(n\right)}\in \left[11,15\right]$ dB), training accuracy converges more slowly. Since, the chaotic signals contain more stochastic features, the fluctuations of the curve and difficulty of training increases. However, for $\sigma_{tr}^{\left(n\right)}\in \left[11,15\right]$ dB, the model can achieve a relatively good accuracy rate, and converge to a small final error.

### 4.2. BER Performance under AWGN Channel

Figure 4 shows the simulated results of the BER performance of the DLCSK, non-coherent DCSK, chaotic switching CSK, and antipodal CSK over the AWGN channel. It is noteworthy that the simulated coherent CSK systems are plotted assuming a perfect chaotic synchronization and only provide benchmark data for evaluation. Antipodal CSK is a chaotic modulation scheme with one basis function that can theoretically achieve the BER performance of binary phase shift keying (BPSK) under AWGN channels. In our simulations, the Logistic map is used for antipodal CSK, such that if “0” is to be sent, a chaotic signal $\overset{˘}{x}$ is transmitted, and if “1” is to be sent, -$\overset{˘}{x}$ is transmitted. The results are also compared with those of chaotic switching, a special case of CSK with two basis functions, in which the transmitted samples are obtained from the Chebyshev and the Logistic maps. The BER curve of the chaotic switching scheme is also simulated assuming a correlation receiver with perfect chaotic synchronization.

The proposed SISO DLCSK scheme shows a gain compared to conventional DCSK in the lower SNRs, when the DLCSK system is trained at relatively lower SNR values (Training SNR $\sigma_{tr}^{\left(n\right)}$∈[11,15] dB). Since the auto and cross correlation properties of the chaotic signals are similar in this experiment, according to Equation (Equation 3), this gain comes from reducing the third term, i.e., $\int_{0}^{T}n\overset{˘}{x}\;dt$. When the training process is performed under different channel conditions, NN can indirectly estimate noise distribution, for use in iterative minimization of the cross-entropy cost function. As an important result, DLCSK shows a more robust behaviour in the test phase.

When $\sigma_{tr}$ is relatively high, the LSTM can only grasp the clean signal. For proper training, the SNR value must help the LSTM learn both clear and noisy samples. The results show that under the AWGN channels, when the $\sigma_{tr}^{\left(n\right)}$∈[19,23] dB, the performance of the SISO DLCSK is close to the conventional DCSK system. Therefore, for better BER performance in high-SNR conditions, the receiver should be trained at higher SNRs. These results seem trustworthy because the above-mentioned training options allow us to design a receiver with a flexible data rate depending upon specific channel conditions.

Figure 5 depicts Monte Carlo simulations of the BER performances obtained from the SISO DLCSK system for β=50 under an AWGN channel. The DLCSK system is trained over a limited SNR range, i.e., [19,23] dB, and tested over the whole SNR range. The other parameters for both case 1 and 2 are selected according to Table 1. The results show that the BER performance of DLCSK has a low sensitivity to changes in the hyper-parameters, such as the number of training samples *S* and the number of hidden units *H*. There is a trade-off between security and classification accuracy. When a system is trained with $\rho_{train}=3.6$, using $\rho_{test}=3.6$ results in more non-periodic behavior. However, reducing the $\rho_{test}$ may result in a better BER performance.

### 4.3. Confusion Matrix, Sensitivity, and Specificity

We provide the confusion matrix of the proposed method, which helps in analyzing the performance of our classification algorithm. Figure 6 depicts an example of two confusion matrices with 10,000 chaotic symbols for AWGN channels for β=50, $\sigma_{tr}^{\left(n\right)}$∈[19,23] dB, and test SNRs = {16,23} dB.

In Figure 7, the sensitivity and specificity measures extracted from the confusion matrix are introduced. Both are statistical measures for the performance of a binary classification test that are widely used in the literature [84,85]. The sensitivity or True Positive Rate (TPR) measures the proportion of Logistic maps that are correctly identified. The specificity or True Negative Rate (TNR) measures the proportion of Chebyshev maps that are correctly identified. The terms "True Positive (TP)”, “False Positive (FP)”, “True Negative (TN)”, and “False Negative” address the correctness of a classification test. For example, if the condition is sending the signals generated by the Logistic map, “TP” means “correctly predicted as Logistic map”, “FP” means “incorrectly predicted as Logistic map”, “TN” means “correctly predicted as Chebyshev map”, and “FN” means “incorrectly predicted as Chebyshev map”.

Figure 8 shows the sensitivity and specificity of different bifurcation parameters. The sensitivity curve in this test shows how capable LSTM can classify samples that are generated by the Logistic maps. Sensitivity can also be referred to as the recall or hit rate [86]. It is the percentage of true Logistic maps out of all samples generated by the Logistic maps (i.e., TP/(TP+FN)). The specificity is a measure of how well the test can identify true Chebyshev maps, which can be expressed as TN/(TN+FP). For high SNRs (i.e., SNR>16 dB), even using a single LSTM-based classifier results in sensitivity and specificity rates >95%. However, there is a trade-off between sensitivity and specificity for low SNRs, where the curves show low sensitivity and high specificity. For example, when SNR=12 dB, $\rho_{train}=3.6$, and $\rho_{test}=3.3$, the sensitivity of a single LSTM-based classifier is 65.69% and its specificity is 99.96%, its false negatives and false positives rates are 34.31% and 0.04%, respectively.

Sensitivity values contain useful information about the receiver’s performance that can be utilized for different goals. Here, we evaluate the effect of the changes in bifurcation parameter on the classifier’s performance. For example, $\rho_{train}=3.6$ and $\rho_{test}=3.3$ leads to a better classification accuracy than other parameter settings for SNR<14 dB. However, with $\rho_{test}=3.3$, the Logistic map shows a periodic behavior and this change can bring negative effect on security.

Figure 9 presents the performance obtained for case 1 using different numbers of training epochs (*E*). The system achieves its peak when E=20. This Figure also measures the sensitivity to the number of epochs. It can be observed that the SISO DLCSK system is relatively robust to the number of epochs. For example, for a target BER performance of $10^{-}{}^{2}$, there is a 1 dB gap between the worst-case and the best-case scenarios.

### 4.4. BER Performance under Multi-Path Rayleigh Fading Channels

Figure 10 illustrates a comparison between the BER performance of the SISO DLCSK and conventional DCSK over multi-path Rayleigh fading channels, for β=50 and $\sigma_{tr}^{\left(n\right)}$∈[19,23] dB. We consider two cases corresponding to different path gain ratios and different path delays. In the first case, a two-path channel is considered (L=2) in which the two paths have similar average power gain. In this case, the average power gain in each path is 0.5, (i.e., $E\left(\alpha_{1}^{2}\right)=E\left(\alpha_{2}^{2}\right)=0.5$), with $\tau_{1}=0$, and $\tau_{2}=2$. In the second case, three paths (L=3) are considered with different average power gains. The average power gains are $E\left(\alpha_{1}^{2}\right)=1/7,E\left(\alpha_{2}^{2}\right)=2/7$, and $E(\alpha_{3}^{2})=4/7$ with $\tau_{1}=0,\tau_{2}=3$, and $\tau_{3}=6$. The average power gain of the third path is 3 dB below the second path, and for the second path it is 3 dB below the first path. When the receiver is already accustomed to chaotic signals, we do not need to transmit a reference signal, it means less energy will be used to transmit one bit. As shown in Figure 10, the SISO DLCSK overcomes the BER inefficiency of the existing DCSK under Rayleigh fading channels. For further benchmark comparisons, we consider the antipodal CSK system. The BER performance of DLCSK is close to antipodal CSK when $E_{b}/N_{0}\leq 10$ dB, and both systems have similar performance when $E_{b}/N_{0}$ is more than 10 dB.

Figure 11 compares the BER performance of the SIMO DLCSK with the SISO-DLCSK under Rayleigh fading channels. The number of receive antennas is M={1,3,5}, and the spreading factor is β=50. The results show that the SIMO DLCSK attains a better BER performance due to the diversity gain. For example, when M=1, to achieve the target $BER=10^{-}{}^{2}$, the required SNR will be about 20 dB. If we increase the number of antennas, i.e., M=5, the required SNR to achieve the $BER=10^{-}{}^{2}$ will only be about 15 dB. Figure 11 also compares the BER performance of the SIMO DLCSK with a recently published LSTM-aided OFDM DCSK in [40]. The LSTM-aided NN calculates the correlations between chaotic modulated OFDM-DCSK signals in order to retrieve the data. For a fair comparison, we consider a SIMO DLCSK with M=5 and an LSTM aided OFDM DCSK with K=5, where *K* indicates the number of independent sub-channels in such a system. The SIMO DLCSK with M=5 independent sub-channels can be fairly compared to LSTM-aided OFDM DCSK with K=5 sub-channels. The LSTM-aided receiver sends reference signals that do not carry useful information. The proposed DLCSK scheme shows an expected gain compared to the LSTM-aided OFDM DCSK because of using indirect channel estimation and reducing the burden of the additional reference sample transmission.

## 5. Conclusions

This paper represents an attempt to develop a new generation of chaos-based communication systems based on DL. We have introduced a trainable DLCSK receiver, which does not need any reference signal transmission or chaotic synchronization. Thus, the DLCSK is more practical, in terms of reliability and compatibility with modern communication infrastructure. A multi-antenna design is presented to achieve a diversity gain. The main objective of the multi-antenna receiver is to improve the classification accuracy of the individual classifiers. Simulation results verify that the SIMO DLCSK system provides an excellent BER performance without channel estimation and complex combining modules at the receiver side. The proposed architecture is an appealing candidate for next generation wireless communications, such as massive MIMO systems and cloud/edge-based communications.

## Figures and Tables

**Figure 1 sensors-22-00333-f001:**
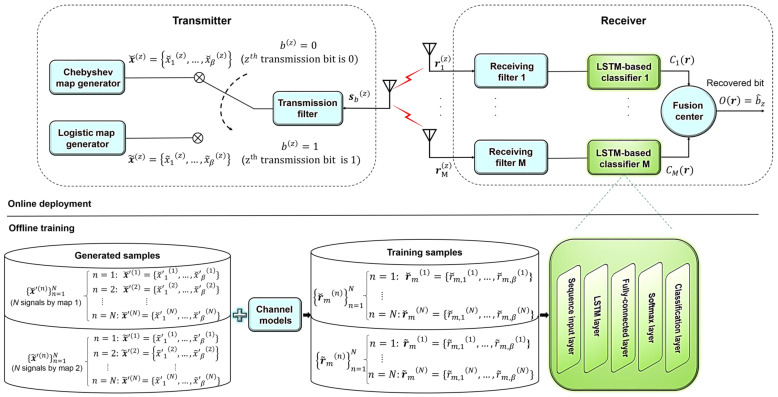
The SIMO DLCSK digital communication system.

**Figure 2 sensors-22-00333-f002:**
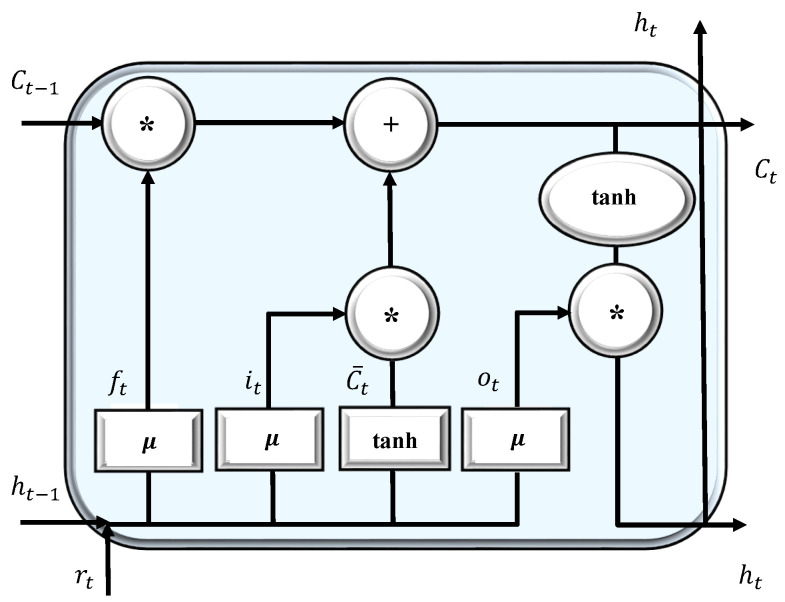
The LSTM cell structure.

**Figure 3 sensors-22-00333-f003:**
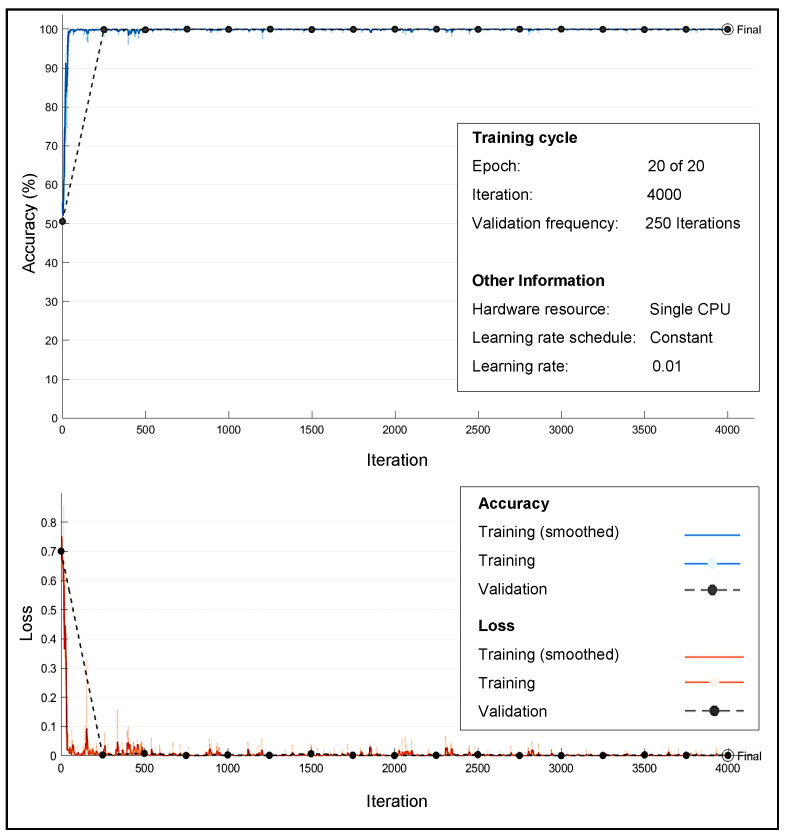
Training metrics when the training SNR $\sigma_{tr}^{\left(n\right)}\in \left[19,23\right]$ dB.

**Figure 4 sensors-22-00333-f004:**
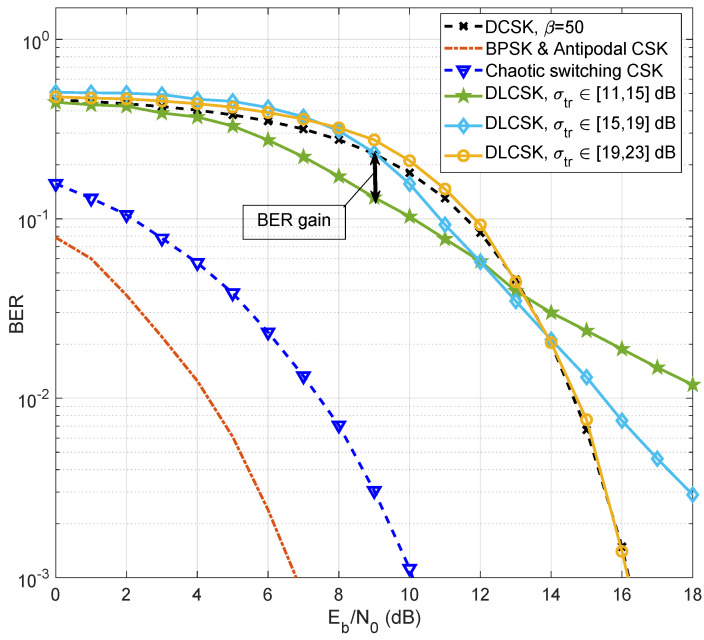
BER curve of the SISO DLCSK under AWGN channel for different $[\sigma_{tr,min}$, $\sigma_{tr,max}]$, β=50, in comparison with the benchmark modulations.

**Figure 5 sensors-22-00333-f005:**
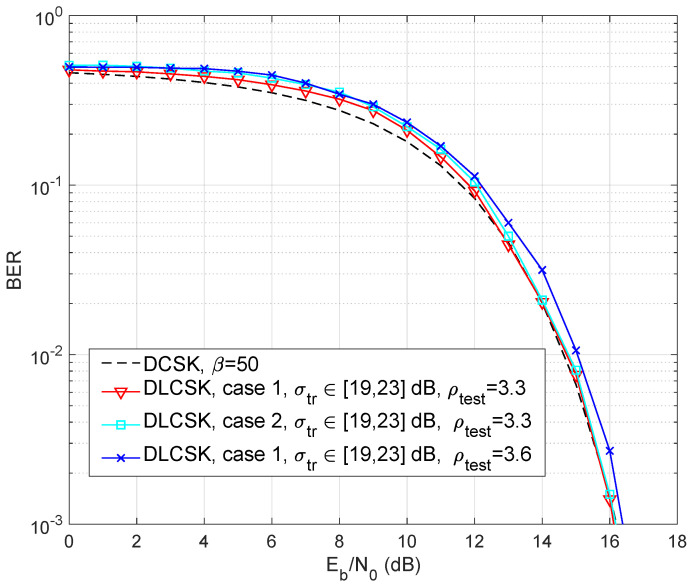
BER curve of the SISO DLCSK under AWGN channel, trained at a random SNR ∈[19,23] dB, and β=50, for case 1, and case 2. There is also the results for $\rho_{test}\in \left\{3.3,3.6\right\}$.

**Figure 6 sensors-22-00333-f006:**
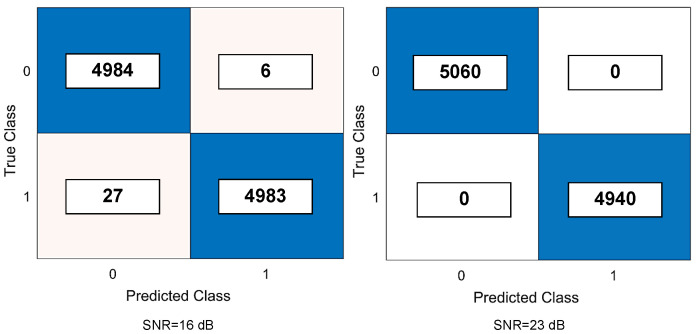
Confusion chart of the LSTM-based classifier under AWGN channel for β=50, $\sigma_{tr}^{\left(n\right)}$∈[19,23] dB, and test SNRs= {16,23} dB.

**Figure 7 sensors-22-00333-f007:**
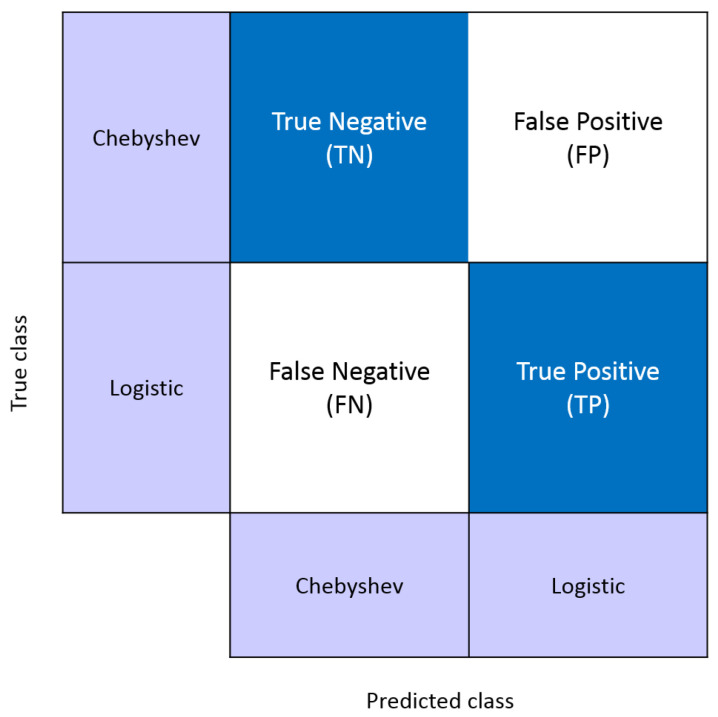
Calculation of sensitivity and specificity from the confusion matrix.

**Figure 8 sensors-22-00333-f008:**
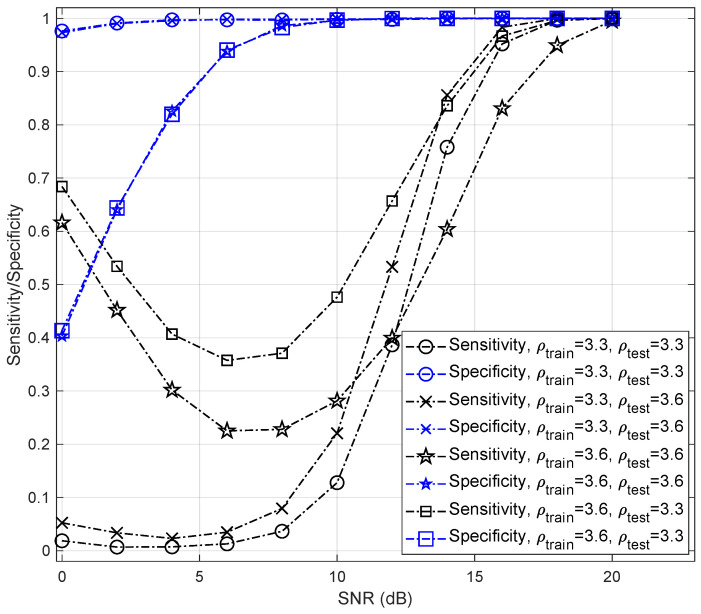
Sensitivity and specificity of LSTM-based classifier under AWGN channel for different values of bifurcation parameters, β=50, and $\sigma_{tr}^{\left(n\right)}$∈[19,23] dB.

**Figure 9 sensors-22-00333-f009:**
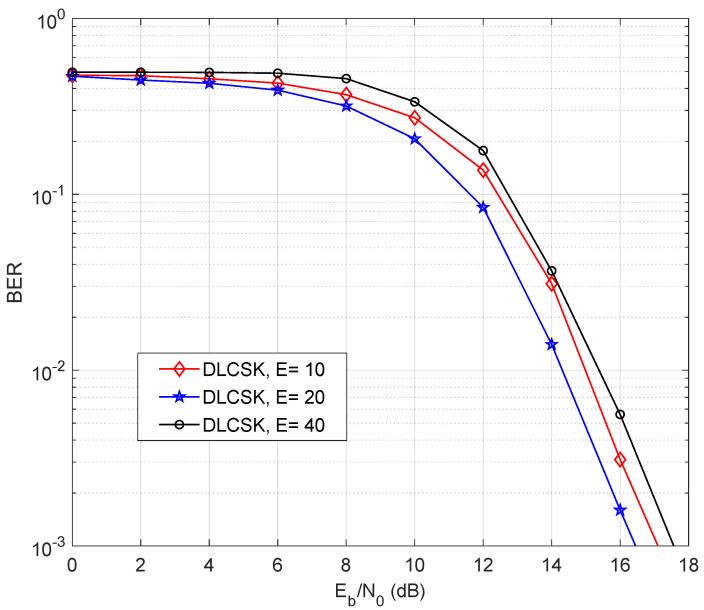
BER of SISO DLCSK under AWGN channel, for β=50 and different numbers of training epochs (E).

**Figure 10 sensors-22-00333-f010:**
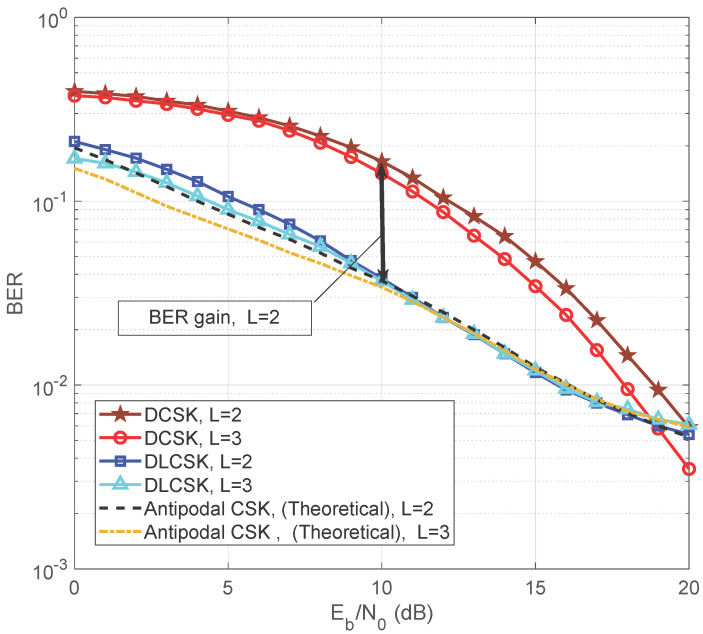
BER of SISO DLCSK for two-path and three-path Rayleigh fading channels, $\sigma_{tr}^{\left(n\right)}$∈[19,23] dB and β=50, in comparison with the benchmark modulations.

**Figure 11 sensors-22-00333-f011:**
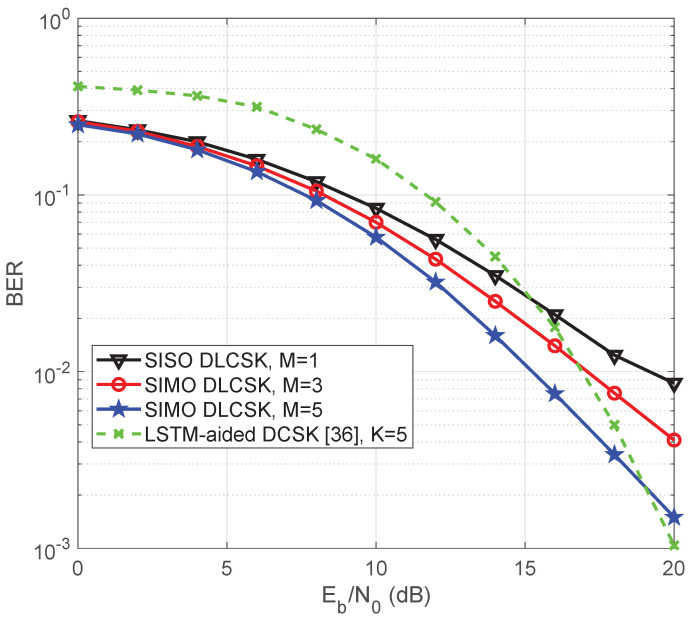
BER curve of the SIMO DLCSK under Rayleigh fading channel, for β=50, in comparison with the LSTM-aided OFDM DCSK [40], for K=5.

**Table 1 sensors-22-00333-t001:** List of the parameters.

Variable	Description	Case 1	Case 2
*S*	Number of training samples (for each class)	$10^{5}$	$10^{4}$
*N*	Number of training signals (for each class)	2000	200
β	Length of chaotic signals	50	50
$w_{v}$	Number of validation samples (for each class)	$3\times 10^{3}$	$10^{3}$
*Z*	Number of test signals	$10^{4}$	$10^{4}$
$a_{t}$	Mini-batch size	200	200
*H*	Number of hidden units	100	10
*E*	Number of epochs	20	500
η	Learning Rate	$10^{-}{}^{2}$	$10^{-}{}^{2}$

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
