# Peer review of "Design of a SIMO Deep Learning-Based Chaos Shift Keying (DLCSK) Communication System"

_sensors, 2022, doi:10.3390/s22010333_

Round 1
Reviewer 1 Report
In this paper, the authors propose an improved Chaos Shift Keying (CSK) demodulation scheme for chaos-based wireless communication systems. The proposed CSK scheme is employing Deep Learning, in order to train offline a Long Short-Term Memory (LSTM)-based receiver in order to be able to classify the modulated signals and retrieve the transmitted messages without the need of chaos synchronization of the conventional CSK scheme or the reference signal transmissions of the Differential CSK (DCSK). The proposed receiver is easily harmonized with existing CSK transmitters achieving remarkable BER performance improvement compared to the conventional DCSK scheme according to the provided simulation results for AWGN and Rayleigh fading channels.
Moreover, a multi-antenna design of the DLCSK receiver that uses an LSTM for each antenna, is also presented, using a fusing method to achieve a diversity gain, where the outputs of all LSTMs are combined using a majority voting strategy. This design maintains the advantages of traditional Multi-Antenna and Chaos-based systems as this architecture is simpler and more efficient for applications where channel estimation is problematic, such as massive MIMO, mmWave, and cloud-based communication systems.
Overall, the paper is well-written, well-structured, and provides a plethora of references to back the authors’ statements. Finally, it presents detailed simulation results to back the authors' claims.
Few minor comments /editing :
1. Line 64-65:
“Some few research analysis have taken out the features of chaotic modulated signals to use in conjunction with DL [33–36].”
The sentence should be rephrased in order to be clearer to the reader. As a minimum, remove the word ”few”.
2. Authors should clarify if the results in figures 4,5 correspond to a classifier trained only at [19,23]dB, while tested down to 0dB, or are an assortment of results from various classifiers trained at their corresponding SNR ranges. The same for figures 10,11.
3. Line 386: A single sentence statement regarding the training metrics for SNR [11,15dB] could be beneficial in order to identify the relative difficulty to train at lower SNRs (since is provided for SNR ~20dB in figure 3).
4. Line 362: Figure 3 instead of “the figure 3”
Reviewer 2 Report
This paper investigates the deep learning (DL)-based Chaos Shift Keying (DLCSK) communication system. The main work of the paper focuses on the design of the DL-based receiver. It is shown that the proposed methods are effective to some degree.
Some of my concerns are as the following.
- The paper mentioned two different types of CSK schemes, i.e., the coherent CSK and the differential CSK. However, the authors should highlight which kind of scheme is the studied scheme for this paper.
- It seems the authors did not explain the rationale for solving the existed drawback in DCSK.
- There is a lack of state-of-the-art DL-based methods in the introduction. For example, the DL-based detection methods, i.e.,
[1] Zha, Xiong, et al. "A deep learning framework for signal detection and modulation classification." Sensors 19.18 (2019): 4042.
[2] C. Liu, Z. Wei, D. W. K. Ng, J. Yuan, and Y.-C. Liang, “Deep transfer learning for signal detection in ambient backscatter communications,” IEEE Trans. Wireless Commun., vol. 20, no. 3, pp. 1624–1638, Mar. 2021.
- For the DL-based channel estimation method, the following methods can also provide satisfactory performance, which should be added for discussion to improve the quality of references.
[1] C. Liu, X. Liu, D. W. K. Ng, and J. Yuan, “Deep Residual Learning for Channel Estimation in Intelligent Reflecting Surface-Assisted Multi-User Communications,” IEEE Trans. Wireless Commun., Aug. 2021 [Early Access], doi: 10.1109/TWC.2021.3100148.
[2] Deep learning-based channel estimation [J]. IEEE Communications Letters, 2019, 23(4): 652-655.
- How to generate the training dataset? Is it available for practical systems?
The hyperparameters of the adopted neural network should be provided for a better understanding of this work.
